# Identification of Prescribing Patterns in Hemodialysis Outpatients Taking Multiple Medications

**DOI:** 10.3390/pharmacy11020043

**Published:** 2023-02-23

**Authors:** Hiroyuki Nagano, Koji Tomori, Mano Koiwa, Shotaro Kobayashi, Masahiro Takahashi, Hideki Makabe, Hirokazu Okada, Akifumi Kushiyama

**Affiliations:** 1Department of Pharmacy, Saitama Medical University Hospital, 38 Morohongo, Moroyama-machi, Iruma-gun, Saitama 350-0495, Japan; 2Department of Pharmacotherapy, Meiji Pharmaceutical University, 2-522-1 Noshio, Kiyose-shi, Tokyo 204-8588, Japan; 3Department of Nephrology, Faculty of Medicine, Saitama Medical University, 38 Morohongo, Moroyama-machi, Iruma-gun, Saitama 350-0495, Japan; 4Department of Pharmacy, Sonoda Daiichi Hospital, 4-1-12 Takenotsuka, Adachi-ku, Tokyo 121-0813, Japan

**Keywords:** multidrug administration, hemodialysis, ATC classification, LCA

## Abstract

We investigated the relationship between multidrug administration and the characteristics, pathophysiology, and drug class in outpatients with hemodialysis. A retrospective cross-sectional study was conducted at Saitama Medical University Hospital in October 2018. Multidrug administration was defined as receiving either more than six drugs or more than the median number of drugs. The drugs used were represented by their anatomical classification codes in the Anatomical Therapeutic Chemistry Classification System (ATC classification). A latent class analysis (LCA) was used to identify clusters at risk of receiving multiple medications. A stepwise logistic regression analysis was performed to select ATC classifications prone to being involved in multidrug administration. As of October 2018, 98 outpatients with hemodialysis were enrolled in the study. In the LCA, when diabetes was the main primary disease, oral hypoglycemic agents available to dialysis patients were limited, but the number of drugs administered was large. Old age, poor nourishment, a long history of dialysis, and chronic nephritis were associated with multidrug administration among nondiabetic patients. In the second level of the ATC classification, the drugs frequently used were coded A02 (drugs for acid-related disorders), A07 (antidiarrheal agents, intestinal anti-inflammatory/anti-infective agents), B01 (antithrombotic agents), and N05 (psycholeptics). The prescribing patterns for either diabetic patients or nondiabetic elderly patients were identified in outpatients with hemodialysis taking multiple medications, and drugs for acid-related disorders, antidiarrheal agents, intestinal anti-inflammatory/anti-infective agents, antithrombotic agents, and psycholeptics are frequently used in those patients.

## 1. Introduction

Polypharmacy, reportedly, increases not only medical costs [1] but also health disorders, such as frailty and fall-related disadvantages [2,3]. The relationship between the drug count and adverse events in elderly patients has been shown to involve the risk of adverse drug events being markedly increased in patients receiving ≥5 drugs [4]. In Japan, polypharmacy is defined as receiving 5–6 drugs, and taking ≥5 drugs increases the risk of falls, while taking ≥6 drugs increases the risk of adverse drug events [5].

Multidrug administration is a risk factor for the progression of chronic kidney disease (CKD) [6], and 91% of CKD patients who are prescribed multiple drugs have a prescription that requires attention to interaction [7]. As CKD progresses, it leads to end-stage kidney disease (ESKD), and hemodialysis is one of the options for renal replacement therapy. The number of dialysis patients (prevalence) per 1 million population is increasing year by year [8]. According to the 2018 United State Renal Data System (USRDS), the prevalence of dialysis patients in Japan is the second highest in the world after Taiwan [9]. In recent years, aging has become a major social problem in Japan, and the function of the kidney declines with age, so the proportion of patients with CKD increases in the elderly [10]. Therefore, it is expected that the number of dialysis patients will continue to increase.

From reports in the United States and Japan, hemodialysis patients are prone to polypharmacy [11,12], and the factors that cause polypharmacy are related to hypertension, diabetes, cardiovascular disease, and dyslipidemia [6]. However, the pattern of drug selection for primary diseases and comorbidities among outpatients with hemodialysis leading to multidrug administration usage is unknown. For pharmacists to intervene in cases of multidrug administration, it is first necessary for them to understand the prescribing patterns.

## 2. Materials and Methods

### 2.1. Objectives

The primary objective was to investigate the relationship between polypharmacy and the characteristics, pathophysiology, and drug classes in hemodialysis outpatients, as the association between drug choice and polypharmacy use in this population is unclear.

### 2.2. Patient Population and Setting

The subjects were outpatient hemodialysis patients at our hospital as of October 2018. We conducted a retrospective cross-sectional study.

### 2.3. Data Collection, Definitions and Outcomes

We collected information on patient background, blood tests, items related to dialysis efficiency and nutritional effects, and drugs used from electronic medical records.

As the characteristics of the study population, the median and interquartile range of the number of drugs used, age, and dialysis history were investigated. The following items were investigated in terms of number and proportion: sex, online-hemodiafiltration (online-HDF), number of visits to other clinical departments, comorbidities (diabetes mellitus, cardiovascular disease, cerebrovascular disease, peripheral arterial disease, liver diseases), and primary diseases of renal failure (diabetic nephropathy, nephrosclerosis, chronic glomerular nephritis, polycystic kidney disease, IgA nephropathy, unknown, other.). The following items were investigated for the median and interquartile range: blood test values (serum albumin, corrected calcium, serum phosphorus, hemoglobin concentration, serum ferritin level, intact parathyroid hormone [i-PTH], β2-microglobulin [M]) and dialysis efficiency/nutritional effect (kt/V, normal Protein Catabolic Rate [nPCR], Geriatric Nutritional Risk Index [GNRI], cardiothoracic ratio [CTR], dry weight [DW]). Multidrug administration was defined as receiving either more than six drugs or more than the median number of drugs.

The items obtained as continuous variables were categorized. Those for which control target values could be defined (corrected calcium level, serum phosphorus level, hemoglobin concentration, serum ferritin level, i-PTH, β2-M, kt/V, nPCR, GNRI) were classified as within or outside the target value. Items that were difficult to define, such as the age, dialysis history, ALB, and DW, were categorized as not less than or less than the median, and the CTR was categorized as not less than or less than 50%. The drugs used were classified using the first level (anatomical group) of the Anatomical Therapeutic Chemical Classification System (ATC classification).

### 2.4. Statistical Analysis

A latent class analysis (LCA) was used to identify the hidden groups of patients with a high prevalence of multidrug administration usage. LCA is a post hoc grouping method [13], and patients with such data as their prescription drugs and primary diseases, indicated as binary data, were classified into three clusters. We next examined the trend in the risk of taking multiple drugs. An LCA assumes that there are latent subgroups (called classes) that affect categorical data that are actually observed and correlate with each other. It is used to analyze the structure by estimating the probability that a subject belongs to a certain class and the conditional probability of the item reaction when the subject belongs to a certain class using the latent class model [14].

A bivariate analysis was performed with the first level of the ATC classification as the explanatory variable for the objective variable, which identified the number of drugs used greater than or equal to the median number of drugs as multidrug use. For classifications that showed a significant difference in this analysis, a logistic regression analysis was performed using the stepwise method for all second-level classifications (treatment method subgroups). Univariate and multivariate logistic regression was performed to evaluate the association between polypharmacy and specific comorbidities (diabetes, cardiovascular diseases cerebrovascular diseases, peripheral arterial diseases, and liver diseases)

The JMP Pro software program, ver. 15.2 (SAS Institute, Cary, NC, USA), was used for these statistical analyses.

## 3. Results

### 3.1. Characteristics of the Study Population

As of October 2018, we had collected 98 outpatients with hemodialysis from our hospital. The characteristics of the study population are shown in Table 1. The median number of drugs was nine. There were 74 and 52 patients who used ≥6 drugs or ≥9 drugs, respectively. The median age was 65 years old, and 52 patients were ≥65 years old. The median dialysis history was 42 months, and the number of patients who had been on dialysis for over 42 months was 49.

There were 32 women, 46 online-HDFs, and 41 in other departments. Diabetes was the most common comorbidity (45 cases), followed by cardiovascular disease (36 cases), cerebrovascular disease (15 cases), liver diseases (14 cases) and peripheral arterial disease (11 cases). Forty-five patients had diabetes, thirty-five were taking diabetes medications, and nineteen were being treated with insulin. Seventeen patients were using one diabetes medication, thirteen patients were using two, four patients were using three, and one patient was using four.

Regarding the primary disease of renal failure, diabetic nephropathy was the most common (38 cases), followed by nephrosclerosis (26 cases), with diabetic nephropathy and nephrosclerosis overlapping in four cases. Chronic glomerulonephritis was noted in 12 cases, polycystic kidney disease in 9, IgA nephropathy in 5, unknown primary disease in 6, and others in 10 cases.

The median values for each blood test are shown: serum albumin 3.5 g/dL, corrected calcium 9.0 mg/dL, serum phosphorus 5.1 mg/dL, hemoglobin concentration 11.3 g/dL, serum ferritin level 87.0 ng/mL, i-PTH 153.4 pg/mL, and β2-M 24.1 mg/L. Serum albumin values were outside the control target range in 48 cases (49.0%), corrected calcium in 15 cases (15.3%), serum phosphorus in 20 cases (20.4%), hemoglobin concentration in 30 cases (30.9%), serum ferritin level in 64 cases (66.7%), i-PTH in 24 cases (24.5%), and β2-M in 21 cases (21.9%). The median values for items related to dialysis efficiency and nutritional effect were kt/V 1.52, nPCR 0.82 g/kg/day, and GNRI 92.33. The kt/V was outside the control target range in 34 cases (35.1%), nPCR in 67 cases (69.1%), and GNRI in 38 cases (40.9%). The CTR was ≥50% in 35 cases (36.5%). The median DW was 59.1 kg, and there were 48 patients weighing ≥59.1 kg.

Among first-level ATC classifications, codes A, C, and V were used by more than 80% of patients.

### 3.2. The LCA

The LCA was performed for 98 outpatients on dialysis, resulting in a 3-class model (Figure 1).

In class 1 (C1), most patients were in the multidrug group of ≥6 drugs, and a substantial number were in the multidrug group of ≥9 drugs. C1 was characterized by diabetes as a comorbidity, and the primary disease was diabetic nephropathy in 90.2% of cases. There were almost no cases of polycystic kidney disease or chronic nephritis. When logistic regression analysis was performed (Appendix A), diabetes was the only comorbidity that was independently and significantly associated with other diseases for multidrug use. Compared with other groups, C1 consulted other clinical departments more frequently, and was often hypocalcemic (19.1%) and anemic (16.0%), but ferritin levels were not decreased. In 90.2% of cases, the nPCR did not meet management goals. The DW tended to be higher than the median. A total of 99.5% of C1 patients used group A drugs (gastrointestinal tract and metabolism), while 87.1% used group C drugs (circulatory system), 47.0% used group N drugs (nervous system), and 47.0% used group B drugs (blood and hematopoietic organs).

In class 2 (C2), 91.4% of the patients were in the multidrug group with ≥6 drugs. C2 was characterized by almost no patients having diabetes as a comorbidity, as these patients tended to have vascular diseases. The primary disease was chronic nephritis, followed by nephrosclerosis. Contrary to C1, high calcium was prominent in C2 (15.2%). C2 also had intact PTH that is high (22.5%) and low (26.1%), falling on both sides of the control target. As in C1, anemia was common, but ferritin was above the control target (15.3%) in many cases. C2 also had the highest percentage of patients with high β2-microglobulin level (30.1%) in the three groups. Most of the drugs used were from groups A (gastrointestinal tract and metabolic action), C (circulatory system), and H (systemic

In class 3 (C3), there were few patients who took ≥6 drugs, and almost none took ≥9 drugs. C3 was characterized by almost no cases of diabetes as a comorbidity or diabetic nephropathy as a primary disease. Renal sclerosis was present. Compared with other clusters, hyperphosphatemia was more common in C3 (28.8%) than in the others. Anemia was rarely seen in C1 (4.1%). C3 patients tended to use fewer drugs in groups B (blood and hematopoietic organs) and N (nervous system) and more in group M (musculoskeletal system). Almost all the patients in C3 were <65 years old and had never visited any other clinical departments. The GNRI was able to be managed to meet appropriate goals.

### 3.3. Results of a Bivariate Analysis (Fisher’s Exact Test)

The relationship between multidrug administration (≥9 drugs per day) and each first-level ATC classification (anatomical group) was examined (Table 2). Significant differences were found among ATC classifications, with A (Alimentary Tract and Metabolism), B (Blood and Blood-forming Organs), C (Cardiovascular System), and N (Nervous System) drugs used significantly more frequently than others.

### 3.4. A Logistic Regression Analysis Using Stepwise Variable Selection

A stepwise regression analysis with multidrug administration as the objective variable was performed using treatment subgroup codes (second category of A, B, C, and N codes), which were significant according to a bivariate analysis (Table 3). A02 (drugs for acid-related disorders), A07 (antidiarrheal agents, intestinal anti-inflammatory/anti-infective agents), B01 (antithrombotic agents), and N05 (psycholeptics) drugs were selected. Furthermore, the use of these drugs was associated with polypharmacy independent of each other as well as with additive polypharmacy.

## 4. Discussion

The number of drugs used by dialysis patients tends to be large. A US study reported that the average number of drugs used per day was 11, and the number of tablets taken was 19 [11], while a Japanese study reported an average of 8.6 oral drugs taken per day, with 17.8 tablets [12]. The present results were similar to those of other studies in Japan. Such multidrug use is attributed to most dialysis patients having end-stage renal disease (ESRD) as well as other common chronic diseases, such as hypertension, diabetes, cardiovascular disease, and mineral and bone disorders [12]. Indeed, almost half of the patients in the present study visited multiple clinical departments, suggesting that they had a number of chronic diseases.

Several aspects concerning the patient image leading to multidrug administration have been clarified. Old age and poor nutritional status have previously been reported as characteristics of patients with polypharmacy [15]. Patients with diabetes as the primary disease tend to show considerably more multidrug use than patients without diabetes. However, the number of oral hypoglycemic agents available for dialysis patients is limited. In general, SU agents, biguanides, and some glinides, which are characterized by renal excretion, are contraindicated. Furthermore, SGLT2 inhibitors are not usually effective [16]. In fact, the number of nonhypoglycemic agents likely contributed to the increase in polypharmacy usage in diabetic patients with dialysis. A cohort study of diabetics estimated that diabetic patients had a 13% higher risk of upper gastrointestinal bleeding than nondiabetic patients [17], and diabetes was a risk factor for gastric ulcers in elderly dialysis patients [18]. *Helicobacter pylori* infection is more common in patients with type 2 diabetes than in nondiabetic patients [19], and it is possible that more drugs for treating gastrointestinal ulcers are used in diabetes patients than in nondiabetes patients as well. In addition, diabetes is a risk factor for the onset of constipation in dialysis patients [20], so laxatives are often used for constipation in dialysis patients with diabetes. Moreover, depressive subjects reportedly have a significantly higher prevalence of diabetes than nondepressive ones [21], so psycholeptics, including sleeping pills, may be used due to sleep disorders caused by depressive symptoms.

One of the backgrounds of patients using antithrombotic drugs is coronary artery disease. About half (53.3%) of CKD patients have significant coronary artery disease, even if they are asymptomatic at the time of dialysis induction, and among CKD patients with diabetes, 83.3% have coronary artery disease [22]. Patients with CKD during the conservative period before dialysis have a significantly higher prevalence of peripheral arterial disease than the general population [23].

Aside from diabetic patients, we also noted that those with high frailty nature, such as the elderly and female [24], were likely to use ≥9 drugs in our study. In a cross-sectional study [15] of elderly people with frailty, the frequency of using ≥10 drugs was significantly higher than in those without frailty. There have also been several studies [25,26,27,28,29] describing the relationship between frailty and polypharmacy. In addition, the group of nondiabetic patients with multidrug administration seen in our study were underweight, had a high malnutritional risk, and had unregulated mineral bone metabolism The association between frail and undernourished patients and polypharmacy indicates that these patients are a key concern for polypharmacy, which is of a different nature than the diabetic group.

We identified the drugs most often used in patients with polypharmacy. To improve or prevent multidrug administration, it is necessary to be aware of the fact that the number of drugs used may increase indirectly due to the medications used in patients with a variety of symptoms such as acid-related disorders, diarrhea, intestinal infection, anxiety, or insomina, even if those medicines are not themselves direct treatments for the primary diseases. The use of acid-related disease therapeutic drugs, such as proton pump inhibitors (PPIs), contributed to multidrug administration.

Several limitations associated with the present study warrants mention. Since the number of target patients was relatively small, the analysis was divided into three arbitrary classes. Therefore, some patient characteristics that should have been extracted may have been overlooked. Another limitation is ethnicity, as our data were mainly obtained from Japanese patients, which may limit the extrapolation of our results to other populations. Finally, because this was a cross-sectional study, what interventions for drug use should be implemented to eliminate polypharmacy should be explored in future prospective studies.

## 5. Conclusions

Outpatients with hemodialysis were found to use a large number of drugs. The prescribing patterns for diabetes patients and nondiabetic elderly patients were identified in outpatients with hemodialysis taking multiple medications, and drugs for acid-related disorders, antidiarrheal agents, intestinal anti-inflammatory/anti-infective agents, antithrombotic agents, and psycholeptics were found to be frequently used in those patients.

## Figures and Tables

**Figure 1 pharmacy-11-00043-f001:**
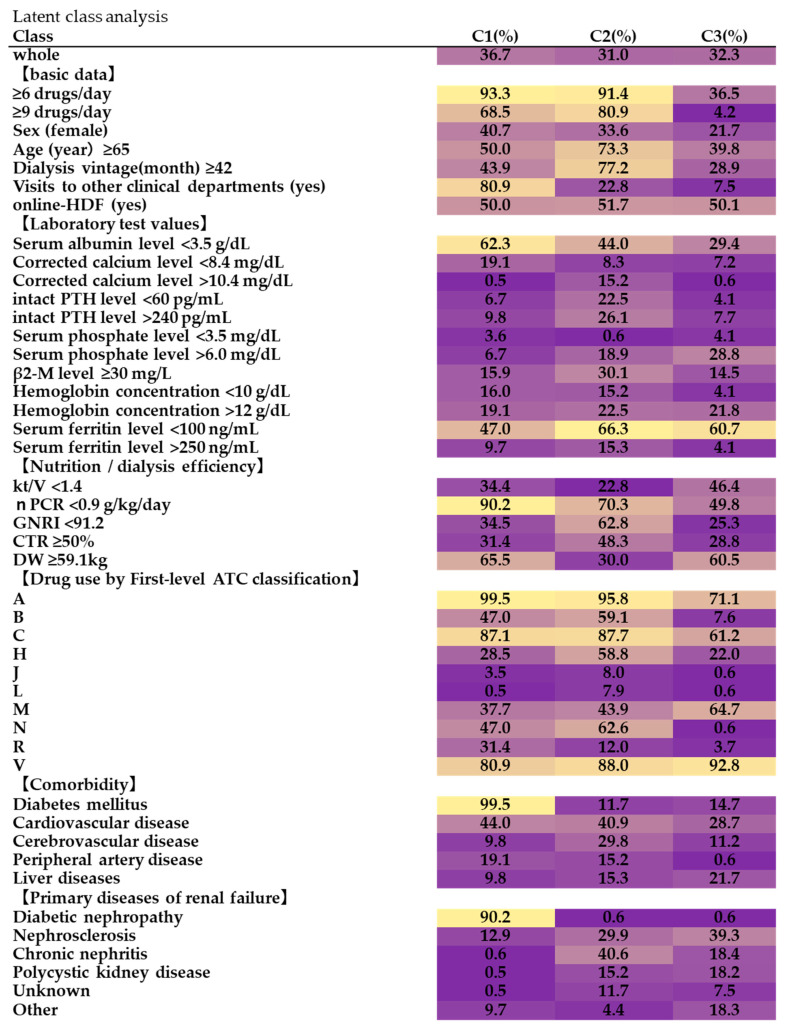
Findings of the latent class analysis (LCA). An LCA was performed to identify the patient clusters with specific profiles and a high prevalence of polypharmacy, resulting in a 3-class model. The values in the figure indicate the frequency at which the criteria were met. Low values are shown in purple, while high values are shown in yellow. Continuous variables were categorized using the thresholds shown in Table 1, such as within or outside the target value or not less than or less than the median. Hormones, excluding sex hormones and insulin). Compared to C3, there were more B (blood and hematopoietic organs) and N (nervous system) drug users. These patients were all ≥65 years old and had a long dialysis history, low DW, and GNRI that was often outside the target range.

**Table 1 pharmacy-11-00043-t001:** Characteristics of the study population.

Number of Patients Analyzed	98
Number of drugs per day (median [IQR])	9 (5.8–11.0)
Number of patients receiving polypharmacy (≥6 drugs/day) (n [%])	74 (75.5)
Number of patients receiving polypharmacy (≥9 drugs/day) (n [%])	52 (53.1)
Age (years) (median [IQR])	65 (56.8–73.0)
≥65 (n [%])	52 (53.1)
Dialysis vintage (month) (median [IQR])	42 (21.8–113.5)
≥42 (n [%])	49 (50.0)
Sex (female) (n [%])	32 (32.7)
online-HDF (yes) (n [%])	46 (46.9)
Visits to other clinical departments (yes) (n [%])	41 (41.8)
**Comorbidity**	
Diabetes mellitus (n [%])	45 (45.9)
Cardiovascular disease (n [%])	36 (36.7)
Cerebrovascular disease (n [%])	15 (15.3)
Peripheral artery disease (n [%])	11 (11.2)
Liver diseases (n [%])	14 (14.3)
**Primary diseases of renal failure**	
Diabetic nephropathy (n [%])	34 (34.7)
Nephrosclerosis (n [%])	26 (26.5)
Chronic nephritis (n [%])	17 (17.3)
Polycystic kidney disease (n [%])	9 (9.2)
unknown (n [%])	6 (6.1)
Other (n [%])	10 (10.2)
**Blood test values**	
Serum albumin level, (g/dL) (median [IQR])	3.5 (3.2–3.7)
<3.5 g/dL (n [%])	48 (49.0)
Corrected calcium level (mg/dL) (median [IQR])	9 (8.7–9.4)
<8.4, >10.4 mg/dL (n [%])	15 (15.3)
<8.4 mg/dL (n [%])	11 (11.2)
>10.4 mg/dL (n [%])	4 (4.1)
Serum phosphate level (mg/dL) (median [IQR])	5.1 (4.5–5.8)
<3.5, >6.0 mg/dL (n [%])	20 (20.4)
<3.5 mg/dL (n [%])	2 (2.0)
>6.0 mg/dL (n [%])	18 (18.4)
intact PTH level (pg/mL) (median [IQR])	153.4 (102.2–212.6)
<60, >240 pg/mL (n [%])	24 (24.5)
<60 pg/mL (n [%])	10 (10.2)
>240 pg/mL (n [%])	14 (14.3)
β2-M level (mg/L) (median [IQR]) (n = 96)	24.1 (20.0–29.0)
≥30 mg/L (n [%])	21 (21.9)
Hemoglobin concentration (g/dL) (median [IQR]) (n = 97)	11.3 (10.8–11.9)
<10, >12 g/dL (n [%])	30 (30.9)
<10 g/dL (n [%])	11 (11.3)
>12 g/dL (n [%])	19 (19.6)
Serum ferritin level (ng/mL) (median [IQR]) (n = 96)	87.0 (53.3–150.5)
<100, >250 ng/mL (n [%])	64 (66.7)
<100 ng/mL (n [%])	54 (56.3)
>250 ng/mL (n [%])	10 (10.4)
**Dialysis efficiency and nutritional effect**	
kt/V (median [IQR]) (n = 97)	1.52 (1.3–1.7)
<1.4 (n [%])	34 (35.1)
nPCR (g/kg/day) (median [IQR]) (n = 97)	0.82 (0.7–0.9)
<0.9 g/kg/day (n [%])	67 (69.1)
GNRI (median [IQR]) (n = 93)	92.33 (88.1–98.0)
<91.2 (n [%])	38 (40.9)
CTR (%) (median [IQR]) (n = 96)	47 (44.3–52.0)
≥50% (n [%])	35 (36.5)
Dry weight (kg) (median [IQR])	59.1 (49.5–70.0)
≥59.1 kg (n [%])	48 (50.0)
First-level ATC classification (anatomical group) (n [%])	
A-Alimentary Tract and Metabolism	88 (89.8)
B-Blood and Blood Forming Organs	35 (35.7)
C-Cardiovascular System	79 (80.6)
H-Systemic Hormonal Preparations, Excl. Sex Hormones and Insulins	33 (33.7)
J-Anti-infectives for Systemic Use	4 (4.1)
L-Antineoplastic and Immunomodulating Agents	2 (2.0)
M-Musculo-Skeletal System	46 (46.9)
N-Nervous System	38 (38.8)
R-Respiratory System	17 (17.3)
V-Various	86 (87.8)

**Table 2 pharmacy-11-00043-t002:** Number of patients receiving polypharmacy (≥9 drugs/day) by first-level ATC classification (anatomical group).

Variable			<9	≥9	Ratio (%)	*p* Value
A	Alimentary Tract and Metabolism	(−)	10	0	0.0	<0.001
		(+)	36	52	59.1	
B	Blood and Blood-forming Organs	(−)	38	25	39.7	<0.001
		(+)	8	27	77.1	
C	Cardiovascular System	(−)	15	4	21.1	<0.001
		(+)	31	48	60.8	
H	Systemic Hormonal Preparations, excl. Sex Hormones and Insulins	(−)	35	30	46.2	0.085
		(+)	11	22	66.7	
J	Anti-infectives for Systemic Use	(−)	46	48	51.1	0.120
		(+)	0	4	100.0	
L	Antineoplastic and Immunomodulating Agents	(−)	46	50	52.1	0.500
		(+)	0	2	100.0	
M	Musculo-Skeletal System	(−)	25	27	51.9	0.842
		(+)	21	25	54.3	
N	Nervous System	(−)	40	20	33.3	<0.001
		(+)	6	32	84.2	
R	Respiratory System	(−)	42	39	48.1	0.059
		(+)	4	13	76.5	
V	Various	(−)	7	5	41.7	0.540
		(+)	39	47	54.7	

**Table 3 pharmacy-11-00043-t003:** Stepwise variable selection in a logistic regression analysis.

Number of Patients Receiving Polypharmacy (≥9 Drugs/Day)			
	AOR	95% Confidence Interval	
		Lower	Upper
A02 (Drugs for Acid-related Disorders)	11.2	3.1	40.6
A07 (Antidiarrheal agents, Intestinal Anti-inflammatory/Anti-infective Agents)	4.9	1.1	22.3
B01 (Antithrombotic Agents)	6.8	1.8	25.4
N05 (Psycholeptics)	10.5	2.3	47.9

## Data Availability

The data were used exclusively for the research conducted as part of this study and were kept confidential.

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
