# Peer review of "Identification of Prescribing Patterns in Hemodialysis Outpatients Taking Multiple Medications"

_pharmacy, 2023, doi:10.3390/pharmacy11020043_

Round 1

Reviewer 1 Report

The present paper deals with analysis of multiple drug administration patterns at patients with hemodyalisis. In a cohort of 98 patients in total, the LCA method was used, and three major classes of patients (C1-prevailing diabetes mellitus, C2-prevailing chronic nephritis, long-term hemodyalisis, elderly and C3- short-time hemodyalisis, younger) were identified. Patients with diabetes mellitus and diabetic nephropathy as a cause of renal failure were identified as most prone to polypharmacy, as the long-term DM is correlated with large number of pathophysiological processes system-wide (vascular issues, neuropathy, acido-basic disorder that reflect to metabolic and respiratory function). On the other hand, high percent of patients with long-term hemodyalisis deal with cardio-vascular or cerebrovascular issues even if they are non-diabetic, resulting in the high number of administered drugs, in prticular at elderly. Interestingly, a positive relation between frailty (elderly, poorly nourished) and polypharmacy has been observed.

The paper is well conceptualized, interesting and deals with an important topics, since multiple drug administration brings particular risk for patients with renal problems, so it is of general interest to avoid unnecessary drugs. The brought conclusions are intuitively expected, systematically confirmed, clear and valuable.

I have following questions:

- In Figure 1 units are missing and the figure is a bit difficult to follow. The legend is scarce. The criteria for quantification {nutrition/dialysis efficiency} and {laboratory test values} as high of low are unclear. Could you improve?

- How would you explain the relation between frailty and multidrug administration? The relation probably arises from the generally poor health condition as the cause of weight loss, rather than from the primary body structure (worse general condition = malnutrition and weight loss = more drugs)? From that point of view, expression “small-in-size” (row 227) probably should be replaces with some more precise?

-In diabetes mellitus, acidosis induces the administration of many drugs, and also the renal failure due to diabetic nephropathy further affects acidobase balance. Could you discuss the problem from the aspect of your findings?

- Were there any patients with known hepatic disorder in your cohort? If any, it would be interesting to mention separately, as this group would be most sensitive to drugs.

Author Response

First of all, I would like to thank you for all of your constructive comments.

1) In Figure 1 units are missing and the figure is a bit difficult to follow. The legend is scarce. The criteria for quantification {nutrition/dialysis efficiency} and {laboratory test values} as high of low are unclear. Could you improve?

Response:

P.7

Missing units in figure1 inserted (%) at the beginning of the class.

The threshold values are indicated after the variable names in Figure 1. We also carefully reconstructed Table 1, since how the thresholds in Figure 1 were chosen came from Table 1; the details of the classification method also used in Table 1 are given in the text, but also outlined in Figure Legend as follows. 

An LCA was performed to identify the patient clusters with specific profiles and a high prevalence of polypharmacy, resulting in a 3-class model. The values in the figure indi-cate the frequency at which the criteria were met. Low values are shown in purple, while high values are shown in yellow. Continuous variables were categorized using the thresholds shown in Table 1, such as within or outside the target value, or not less than or less than the median.

Since the LCA analysis was conducted again after reflecting the reviewer's remarks, there are slight differences in the figures included.

2) How would you explain the relation between frailty and multidrug administration? The relation probably arises from the generally poor health condition as the cause of weight loss, rather than from the primary body structure (worse general condition = malnutrition and weight loss = more drugs)? From that point of view, expression “small-in-size” (row 227) probably should be replaces with some more precise?

Response:

As you pointed out, from the previous literature that showed the predictive factors of frailty in dialysis patients, the points that should be noted from the results of this time were the elderly and female, so the description was revised accordingly and the citation was added.

Aside from diabetic patients, we also noted that those with high frailty nature, such as the elderly, female [24], were likely to use ≥9 drugs in our study. In a cross-sectional study [15] of elderly people with frailty, the frequency of using ≥10 drugs was significantly higher than in those without frailty. There have also been several studies [25-29] describing the relationship between frailty and polypharmacy. In addition, the group of non-diabetic patients with multidrug administration seen in our study were underweight, had a high malnutritional risk, and had unregulated mineral bone metabolism The association between frail and undernourished patients and polypharmacy indicates that these patients are a key concern for polypharmacy, which is of a different nature than the diabetic group.

3) In diabetes mellitus, acidosis induces the administration of many drugs, and also the renal failure due to diabetic nephropathy further affects acidobase balance. Could you discuss the problem from the aspect of your findings?

Response:

Thank you for pointing out this important point. Diabetic nephropathy can cause acidosis and acid-base imbalance, which can affect the number of drugs used. However, since these are corrected by dialysis after the introduction of dialysis, there was no increase in the number of drugs related to acidosis in this study. I tried to include this in the discussion, but thought it was unlikely to be our focus of the study.

4) Were there any patients with known hepatic disorder in your cohort? If any, it would be interesting to mention separately, as this group would be most sensitive to drugs.

Response:

As indicated, complications of liver disease were investigated. Results showed that this comorbidity was not associated with polypharmacy.

Reviewer 2 Report

Kushiyama et al. conducted a retrospective cross-sectional study in order to identify prescription patterns in HD patients. This study demonstrated the presence of 3 prescribing patterns:
Class 1 pts were more likely to have diabetes and to take A and C group drugs; in Class 2 there was a higher prevalence of vascular disease, while C3 pts were younger. C1 and C2 pts were more likely to take multiple medications.
Moreover, administration of acid-related disorders drugs, intestinal drugs, antithrombotics and psycoleptics were frequently associated with receiving more than 9 drugs/day.

The paper has the merit of focusing on a particular population (HD pts), and the topic is arguing.
However, in order to derive clinical meaning from this study, I suggest to:
-          Expand comorbidities range (e.g. mineral bone disorders, neoplasm ..) if possible.
-          Perform univariate and multivariate logistic regression to assess the association between polypharmacy and a specific comorbidity
-          Add haemoglobin and ferritin to blood test values since the administration of erythropoiesis-stimulating agents and iron supplementation is extremely frequent in HD pts.
-          P values should be shown in multivariate analysis.

Author Response

First of all, I would like to thank you for all of your constructive comments.

Reviewer 2:

1) Expand comorbidities range (e.g. mineral bone disorders, neoplasm ..) if possible.

Response:

The mineral bone abnormalities you mentioned as examples were treated in almost all cases, and no patients were treated for malignant neoplasms, making them an inappropriate factor to explore in relation to polypharmacy. After further review, only liver disease was added to the data. However, since phosphorus and calcium metabolism are clearly associated with differences in the C1-C3 patient picture, the following text was added to Discussion.

P9

In addition, the group of non-diabetic patients with multidrug administration seen in our study were underweight, had a high malnutritional risk, and had unregulated mineral bone metabolism.

2) Perform univariate and multivariate logistic regression to assess the association between polypharmacy and a specific comorbidity

Response:

We performed the analysis indicated for comorbidities. The results are shown in Supplementary Table 1, and we believe that the importance of diabetes is now clear and more clearly discussed. Thank you for your important remarks.

We added sentences as follows:

P3 L115

Univariate and multivariate logistic regression was performed to evaluate the association between polypharmacy and specific comorbidities (diabetes, cardiovascular diseases cer-ebrovascular diseases, peripheral arterial diseases, and liver diseases)

P4 L160

When logistic regression analysis is performed(supplementary table 1), diabetes is the only comorbidity that is independently and significantly associated with other diseases for multidrug use.

3) Add haemoglobin and ferritin to blood test values since the administration of erythropoiesis-stimulating agents and iron supplementation is extremely frequent in HD pts.

Response:

When hemoglobin and ferritin levels were taken into account, C1 and C2 showed more anemia but no decrease in ferritin compared to C3. Although the association with polypharmacy is not clear with regard to anemia-related drug use, we believe that the characteristics of the population are more revealing. We would like to thank the authors for their important suggestions.

Since the LCA analysis was conducted again after reflecting the reviewer's remarks, there are slight differences from before in the figures included.

We added sentence about anemia as follows

P4

Compared with other groups, C1 consulted other clinical departments more frequently, and is often hypocalcemic (19.1%) and anemic (16.0%), but ferritin levels are not decreased. In 90.2% of cases, the nPCR did not meet management goals. The DW tended to be higher than the median. A total of 99.5% of C1 patients used group A drugs (gastrointestinal tract and metabolism), while 87.1% used group C drugs (circulatory system), 47.0% used group N drugs (nervous system), and 47.0% used group B drugs (blood and hematopoietic organs).

In class 2 (C2), 91.4% of the patients were in the multidrug group with ≥6 drugs. C2 was characterized by almost no patients having diabetes as a comorbidity, as these patients tended to have vascular diseases. The primary disease was chronic nephritis, followed by nephrosclerosis. Contrary to C1, high calcium is prominent in C2 (15.2%). C2 also has intact PTH that is high (22.5%) and low (26.1%), falling on both sides of the control target. as in C1, anemia is common, but ferritin is above the control target (15.3%) in many cases. C2 also has the highest percentage of patients with high β2-microglobulin level (30.1%) in the three groups. Most of the drugs used were from groups A (gastrointestinal tract and metabolic action), C (circulatory system), and H (systemic hormones, excluding sex hormones and insulin). Compared to C3, there were more B (blood and hematopoietic organs) and N (nervous system) drug users. These patients were all ≥65 years old and had a long dialysis history, low DW, and GNRI that was often outside the target range.

In class 3 (C3), there were few patients who took ≥6 drugs, and almost none took ≥9 drugs. C3 was characterized by almost no cases of diabetes as a comorbidity or diabetic nephropathy as a primary disease. Renal sclerosis was present. Compared with other clusters, hyperphosphatemia is more common in C3 (28.8%) than in the others. Anemia was rarely seen in C1(4.1%).

4) - P values should be shown in multivariate analysis.

Response:

In studies where randomness cannot be assumed, such as observational studies, interval estimates are preferable to tests using P-values. In this case, it has been advanced in the field of epidemiology that the notation of the P value is unnecessary, and we described our documents according to its recommendation.

https://www.ncbi.nlm.nih.gov/pmc/articles/PMC4877414/

Reviewer 3 Report

The authors of the manuscript “investigation of prescribing patterns in hemodialysis outpatients taking multiple medications“ studied prescription patterns in a retrospective cohort of hemodialysis patients using a latent class analysis in order to identify patients at risk for polypharmacy. The used methods allowed to define different classes of patients with different medication patterns. Statistics and discussion is conclusive although there is a need for language editing. The originality of the study is average.

Author Response

First of all, I want to thank you for appreciating our work.

The document was thoroughly checked again by the document proofreader. A certificate has been submitted. Based on the suggestions of other reviewers, we proceeded with the revision process.

Round 2

Reviewer 2 Report

I have no further comments.